# Expression of GEX1 Orthologs of *Brassica rapa* and *Oryza sativa* Rescued the Nuclear Fusion Defect of the Arabidopsis GEX1 Mutant

**DOI:** 10.3390/plants11141808

**Published:** 2022-07-08

**Authors:** Ayaka Yabe, Shuh-ichi Nishikawa

**Affiliations:** 1Graduate School of Science and Technology, Niigata University, Niigata 950-2181, Japan; tulip.617.ayaka@gmail.com; 2Faculty of Science, Niigata University, Niigata 950-2181, Japan

**Keywords:** nuclear fusion, female gametogenesis, fertilization, membrane fusion, *Arabidopsis thaliana*, *Brassica rapa*, *Oryza sativa*

## Abstract

Nuclear fusion is required for the sexual reproduction of various organisms, including angiosperms. During the life cycle of angiosperms, nuclear fusion occurs three times: once during female gametogenesis, when the two polar nuclei fuse in the central cell, and twice during double fertilization. Nuclear fusion in plant reproduction is achieved by sequential nuclear fusion events: outer and inner nuclear membrane fusion. *Arabidopsis* gamete expressed 1 (GEX1) is a nuclear membrane protein of gametes that is required for nuclear fusion during reproduction. Although orthologs of GEX1 have been identified in various land plants, sequence identities are not high, even between angiosperm GEX1 orthologs; the sequence identity between *Arabidopsis* GEX1 and *Oryza sativa* GEX1 ortholog is lower than 50%. Here, we found that the expression of GEX1 orthologs of *O. sativa*, as well as of *Brassica rapa* from the *Arabidopsis GEX1* promoter, rescued the polar nuclear fusion defect of the *gex1* mutant. We also found that the expression of these GEX1 orthologs rescued the lethality of the *gex1* homozygous mutant, which is proposed to be caused by the sperm nuclear fusion defects upon fertilization. Our results indicate a functional conservation between *Arabidopsis* and *O. sativa* GEX1 orthologs, despite their relatively low sequence identities.

## 1. Introduction

The fertilization of angiosperms consists of multiple processes, starting with the adhesion of pollen grains on the stigma of a pistil. Pollen germinates and grows a tube in the pistil to deliver sperm cells to the female gametophytes in the ovary. After arrival at the female gametophyte, two sperm cells are released from the pollen tube to fertilize two female gametes, the egg and central cell. Fertilization of the egg and central cell results in the formation of the embryo and the endosperm, respectively [1]. Plants have mechanisms that enable the progression of these fertilization processes with high fidelity.

Nuclear fusion is required for the sexual reproduction of various organisms, including angiosperms. During the life cycle of angiosperms, nuclear fusion occurs three times. Two of these events are sperm nuclear fusion during double fertilization. The third nuclear fusion occurs during the development of female gametophytes [2,3]. In most angiosperms, including *Arabidopsis thaliana*, a single megaspore produced by meiosis undergoes three rounds of mitosis. After the second mitosis, a four-nucleate female gametophyte is produced. Cellularization starts after the third mitosis, resulting in the formation of a seven-celled female gametophyte, which consists of one egg cell, one central cell, two synergid cells, and three antipodal cells. The central cell contains two polar nuclei that fuse before pollination to form the secondary nucleus in *Arabidopsis* and other species [2,3].

In contrast to mammalian fertilization, in which nuclear fusion takes place during the first mitotic division by nuclear envelope breakdown and reconstitution, nuclear fusion in plant reproduction is achieved by sequential nuclear membrane fusion events: outer and inner nuclear membrane fusion [4]. Our analyses showed that, in *Arabidopsis thaliana*, the immunoglobulin-binding protein (BiP), a molecular chaperone of the heat shock protein 70 (Hsp70) family in the endoplasmic reticulum (ER), is required for the fusion of polar nuclei [5]. BiP functions in the fusion of outer and inner nuclear membranes of polar nuclei with its regulatory partners, ER-resident J-domain-containing proteins [6]. BiP and ER-resident J-proteins are also involved in sperm nuclear fusion after fertilization. Analyses of the development of the mutant seeds showed that fertilization-coupled sperm nuclear fusion is critical for proper endosperm proliferation [7].

We also showed that *Arabidopsis* gamete-expressed 1 (AtGEX1) is a nuclear membrane protein necessary for nuclear fusion during *Arabidopsis* reproduction [8]. The *AtGEX1* gene was identified by screening genes that are expressed in the sperm cells [9]. The *gex1-1* mutant was isolated from a collection of mutants constructed by the insertion of a T-DNA that expresses β-glucuronidase (GUS) from the pollen-specific *LAT52*-promoter [10,11,12]. Analyses of the *gex1-1* mutant show that AtGEX1 is involved in polar nuclear membrane fusion after fusion of the outer nuclear membrane. AtGEX1 is also involved in sperm nuclear fusion in the fertilized egg and central cells. The resulting *gex1-1* homozygous mutant seeds showed aberrant endosperm proliferation and defects in early embryo development, which resulted in seed abortion [8,13].

Orthologs of AtGEX1 have been identified in various land plants [8]. GEX1 family proteins are part of Kar5/GEX1/Brambleberry family, which includes budding yeast Kar5 and zebrafish Brambleberry proteins [14]. Yeast Kar5 protein is a nuclear membrane protein that is essential for nuclear fusion during yeast mating [15]. Zebrafish Brambleberry is essential for pronucleus fusion in the zygote and for karyomere fusion during early embryogenesis [16]. All land plant GEX1 orthologs contain the Cys-rich domain (CRD) in their N-terminus [14], followed by two or three putative coiled-coil regions and three transmembrane domains (Figure 1A). However, sequence similarities are not high, even between angiosperm GEX1 orthologs [8]. Similar amino acid sequence divergence between angiosperm orthologs was reported in the case of HAPLESS2 (HAP2)/GENERATIVE CELL-SPECIFIC1 (GCS1) protein. The *Oryza sativa* (rice) ortholog of HAP2/GCS1 is 59% identical to *Arabidopsis* HAP2/GCS1 in the N-terminal region (residue 1-559 of *A. thaliana* HAP2/GCS1) and 37% identical at the C-terminus (residue 582-705 of *A. thaliana* HAP2/GCS1). However, the expression of the *O. sativa* ortholog of HAP2/GCS1 failed to rescue the *Arabidopsis hap2-1* fertilization defect [17]. Analysis using interspecific HAP2/GCS1 chimeras showed that amino acid sequence divergence in the N-terminal extracellular domain caused the functional differences between HAP2/GCS1 orthologs of distantly related species.

Here, we asked whether differences in amino acid sequences between angiosperm GEX1 orthologs cause their functional differences in nuclear fusion. We found that the expression of GEX1 orthologs of *Brassica rapa*, which belongs to the same family as *A. thaliana*, (BrGEX1) and *O. sativa* (OsGEX1) from the *AtGEX1* promoter, rescued the polar nuclear fusion defect of the *gex1-1* mutant female gametophytes. We also found that the expression of BrGEX1 and OsGEX1 rescued the lethality of the *gex1-1* homozygous plants, including the seed abortion phenotype, which was suggested to be caused by the sperm nuclear fusion defect at fertilization. These results indicate conserved nuclear fusion activities between *Arabidopsis* and rice GEX1 orthologs, despite their relatively low sequence identities.

## 2. Results

### 2.1. Expression of BrGEX1 and OsGEX1 Rescued the Polar Nuclear Fusion Defect of the gex1-1 Female Gametophytes

Multiple sequence alignments obtained using Clustal Omega [18] showed that the amino acid identities between AtGEX1 and GEX1 orthologs of BrGEX1 and OsGEX1 are 73.8% and 43.8%, respectively (Appendix A). Although the CRD is a region with relatively high amino acid residue conservation, the amino acid identity between AtGEX1 and OsGEX1 was 50.0% (Figure 1A). To analyze the functional differences between angiosperm GEX1 orthologs in nuclear fusion, we examined whether the expression of BrGEX1 and OsGEX1 rescued the nuclear fusion defect of the *Arabidopsis gex1-1* mutant. We first analyzed whether the expression of GEX1 orthologs of BrGEX1 or OsGEX1 rescued the polar nuclear fusion defect of the *gex1-1* mutant female gametophytes. An analysis of mature-stage female gametophytes by confocal laser-scanning microscopy showed that approximately 50% of female gametophytes of *gex1-1*/+ plants contained unfused polar nuclei (Figure 1B–D). As reported previously, the expression of AtGEX1 from the *AtGEX1*-promoter rescued the polar nuclear fusion defects of the *gex1-1* mutant female gametophytes [8]. More than 90% of the ovules of the *gex1-1*/+ line homozygous for the construct containing *AtGEX1* cDNA, driven by the *AtGEX1* promoter (*pAtGEX1*: *AtGEX1*), contained a single secondary nucleus. We introduced constructs containing *BrGEX1* cDNA or *OsGEX1* cDNA driven by the *AtGEX1* promoter (*pAtGEX1*: *BrGEX1* or *pAtGEX1*: *OsGEX1*) into the *gex1-1*/+ plants. Three independent transgenic *gex1-1*/+ lines that were homozygous for *pAtGEX1*: *BrGEX1* and two independent transgenic *gex1-1*/+ lines homozygous for *pAtGEX1*: *OsGEX1* were isolated. More than 90% of the ovules contained a single secondary nucleus in all transgenic lines (Figure 1D). These results indicate that the expression of BrGEX1 or OsGEX1 from the *AtGEX1* promoter rescued the polar nuclear fusion defect of the *gex1-1* mutant female gametophytes.

### 2.2. Expression of BrGEX1 and OsGEX1 Rescued the Lethality of the gex1-1 Homozygous Mutant

By genotyping the progenies of the transgenic *gex1-1*/+ lines homozygous for *pAtGEX1*: *AtGEX1*, *pAtGEX1*: *BrGEX1*, or *pAtGEX1*: *OsGEX1*, we obtained plants that were homozygous for the *gex1-1* mutation. Real-time quantitative PCR analyses showed that the expression of the *AtGEX1* transcripts was not detectable in flowers of the *gex1-1* homozygous plants that were homozygous for the *pAtGEX1*: *BrGEX1* or *pAtGEX1*: *OsGEX1* transgenes (Figure 2A). The *BrGEX1* or *OsGEX1* transcripts were expressed from the introduced transgenes in the isolated transgenic lines, although their expression levels differed among the lines (Figure 2B,C). These results indicate that the expression of BrGEX1 or OsGEX1 from the *AtGEX1* promoter rescued the lethality of the *gex1-1*/*gex1-1* homozygous plants. Since the T-DNA element used for the generation of the *gex1-1* mutant contained a *LAT52*: *GUS* pollen-specific reporter gene, pollen carrying the *gex1-1* allele can be identified using assays for GUS activity. The *gex1-1* mutant is homozygous for the *quartet1* (*qrt1*) mutation, which generates four microspores derived from a single pollen mother cell. These remain attached to each other [19]. While pollen tetrads of the *gex1-1*/+ contained two GUS-positive and two GUS-negative pollen grains, the obtained *gex1-1* homozygous plants produced pollen tetrads containing four GUS-positive pollen grains (Figure 2D–K), confirming that the plants were homozygous for *gex1-1*.

### 2.3. Expression of BrGEX1 and OsGEX1 Rescued Abortion of the gex1-1 Homozygous Mutant Seeds

AtGEX1 is also required for sperm nuclear fusion after fertilization. Sperm nuclear fusion events were defective in the fertilized egg and central cell after the fertilization of *gex1-1* female gametophytes by *gex1-1* pollen. The resulting *gex1-1* homozygous mutant seeds were lethal and showed defects in early embryo development [8]. In contrast, wild-type plants produced siliques with a full seed set (Figure 3A), self-crossed *gex1-1*/+ produced siliques containing approximately 23% aborted seeds, which is statistically consistent with 25% seed abortion (Figure 3B arrowheads and Figure 3F). The expression of AtGEX1 from the *AtGEX1* promoter rescued the seed abortion of the homozygous *gex1-1* mutant. Homozygous *gex1-1* lines that were homozygous for *pAtGEX1*: *AtGEX1* were obtained from three independent *pAtGEX1*: *AtGEX1 gex1-1*/+ transgenic lines [8]. These lines produced siliques with full seed sets, comparable to those observed in wild-type plants (Figure 3C,F). The expression of BrGEX1 or OsGEX1 from the *AtGEX1* promoter also rescued the abortion of the *gex1-1* homozygous mutant seeds. The *gex1-1* homozygous plants that were homozygous for the *pAtGEX1*: *BrGEX1* or *pAtGEX1*: *OsGEX1* transgenes also produced siliques with full seed sets (Figure 3D–F). Since sperm nuclear fusion defects at fertilization were suggested to cause the seed abortion [7,8], these results suggest that the expression of BrGEX1 or OsGEX1 rescued the sperm nuclear fusion defect of the *gex1-1* homozygous mutant seeds.

### 2.4. Structural Similarities between GEX1 Orthologs

The amino acid sequence of OsGEX1 was 43.8% identical to that of AtGEX1. AtGEX1 and OsGEX1 are predicted to contain two and three putative coiled-coil regions, respectively. Despite these differences in their primary structure, the above results show that OsGEX1 has similar nuclear fusion activities to AtGEX1. This is probably due to the relatively high amino acid sequence similarities between AtGEX1 and OsGEX1: their amino acid similarity is higher than 80% in the CRD and coiled-coil regions (Figure 1A and Appendix A). The observed high amino acid sequence similarities could result in conservation of the tertiary structures between AtGEX1 and OsGEX1. Structural predictions by AlphaFold2 [20] showed that OsGEX1 had a similar tertiary structure to that of AtGEX1 (Figure 4). All GEX1 family proteins contain the CRD in the N-terminal region, which is characterized by the presence of six conserved Cys residues [8,14]. Structural prediction showed the CRD to be a globular domain (Figure 4). Following the CRD, the coiled-coil regions of AtGEX1 and OsGEX1 were predicted to form an anti-parallel, α-helical, coiled-coil hairpin. Using this coiled-coil hairpin, the CRD is predicted to be positioned close to the transmembrane domains in both structures. AtGEX1 orthologs contain the putative N-terminal signal sequence (Figure 1A). The predicted structures of AtGEX1 and OsGEX1 lacking their putative signal sequence are similar to those of the full-length proteins (not shown). The observed similarity between the predicted AtGEX1 and OsGEX1 structures can explain the nuclear fusion activities of OsGEX1 in *Arabidopsis*, despite the relatively low identities in their amino acid sequences.

## 3. Discussion

AtGEX1 is a nuclear membrane protein required for all three nuclear fusion events in *Arabidopsis* reproduction, polar nuclear fusion during female gametogenesis and sperm nuclear fusion during double fertilization [8]. GEX1 orthologs have been identified in various land plants. In this study, we showed that the expression of OsGEX1 as well as BrGEX1 rescued the polar nuclear fusion defect of the *gex1-1* female gametophytes. We also obtained the *gex1-1* homozygous plants in the transgenic *gex1-1*/+ lines expressing BrGEX1 and OsGEX1. Since the sperm nuclear fusion defects upon fertilization of the *gex1-1* homozygous seeds are suggested to cause seed abortion [8], our results strongly suggest that the expression of BrGEX1 and OsGEX1 also rescued the sperm nuclear fusion defect of the *gex1-1* mutant. Therefore, OsGEX1 as well as BrGEX1 most likely shares conserved nuclear fusion activities with AtGEX1 in the nuclear fusion events of reproduction.

Rescue of the nuclear fusion defects in the *gex1-1* mutant by the expression of OsGEX1 was probably due to the conserved tertiary structure of AtGEX1 and OsGEX1. The N-terminal region of these proteins is predicted to be consisted of two domains, the CRD and the coiled-coil domain. Since coiled-coils are found in proteins functioning in membrane fusion [21], AtGEX1 and its orthologs possibly function in the membrane fusion process in nuclear fusion. The CRD is found in all GEX1 orthologs [14]. The CRD most likely plays key roles in the nuclear fusion activities of GEX1 orthologs. In addition to the conserved Cys residues, CRD contains amino acid residues conserved between angiosperm GEX1 orthologs [8]. Mutational analysis of these conserved residues would provide insights into AtGEX1 functions in nuclear fusion.

In *Arabidopsis*, fusion of the polar nuclei is completed during female gametogenesis. In contrast, in *B. rapa* and rice, fusion starts during female gametogenesis but is not completed before fertilization [22,23]. Electron microscopy showed that, in rice mature female gametophytes, the partially fused polar nuclei were connected by bridges between the two nuclei and their nucleoplasm was continuous [22], indicating that nuclear fusion proceeded after the fusion of inner nuclear membrane. This is in good contrast to the *gex1-1* mutant female gametophytes, which were defective in the nuclear membrane fusion step following the fusion of the outer nuclear membranes [8]. The completion of polar nuclear fusion in the *gex1-1* female gametophytes expressing BrGEX1 or OsGEX1 supports that the variation in the times of nuclear fusion completion is not due to differences in the activities of GEX1 orthologs. Mechanisms regulating the fusion of polar nuclei after the membrane fusion process, which allow for the completion of polar nuclear fusion during female gametogenesis, may exist in *Arabidopsis* but not in *B. rapa* or rice female gametophytes. Molecular genetics would reveal the mechanisms of functioning in the final steps of polar nuclear fusion.

## 4. Materials and Methods

### 4.1. Plant Materials and Growth Conditions

*Arabidopsis thaliana* qrt1-2 (CS8846; Col-3) was used as the wild type. The *gex1-1* (CS817261; Col-3) mutant allele has been described [13]. Seeds of rapid-cycling *Brassica rapa* (Fast plants) [24] were purchased from In The Woods Group (Aomori, Japan). The seeds were surface-sterilized and sown on soil or Murashige and Skoog (MS) medium (Fuji Film Wako, Osaka, Japan) containing 0.7% agar and 1% sucrose. The plants were grown at 22 °C under continuous light.

### 4.2. Plasmid Construction and Plant Transformation

A plasmid for expression of *Arabidopsis thaliana GEX1* from the *AtGEX1* promoter was previously described [8]. Primers used for plasmid construction are listed in Appendix A. Total RNA was prepared from flowers of rapid-cycling *Brassica rapa* plants using the RNeasy Plant Mini Kit (Qiagen, Tokyo, Japan) and reverse-transcribed to cDNA using the SuperScriptIII first-strand synthesis system (Thermo Fisher Scientific K.K., Tokyo, Japan) according to the manufacturer’s protocol. A cDNA fragment for *Brassica rapa GEX1* (*BrGEX1*) was PCR-amplified using primer set BrGEX1F1/BrGEX1R1 and first-strand cDNA as a template. The amplified 1.8 kb DNA fragment was cloned into the pENTR/D-TOPO vector to generate pSNA139. A cDNA fragment for *Oryza sativa GEX1* (*OsGEX1*) was PCR-amplified using primer set OsGEX1F2/OsGEX1R2 and first-strand cDNA of *Oryza sativa japonica* (Nipponbare) spikelet (a gift from Daisuke Maruyama, Yokohama City University) as a template. The amplified 2.0 kb DNA fragment was cloned into the pENTR/D-TOPO vector to generate pSNA141. The *BrGEX1* and *OsGEX1* constructs from pSNA139 and pSNA141, respectively, were introduced into pSNA136 [8] using LR clonase II (Thermo Fisher Scientific K.K., Tokyo, Japan).

To generate transgenic lines, *Agrobacterium tumefaciens* strain GV3101 was introduced into *Arabidopsis* plants using the floral-dip method [25]. The transgenic plants were selected on MS agar plates containing 50 µg/mL hygromycin and were subsequently transferred to soil.

### 4.3. Microscopy

The ovules were prepared for confocal laser-scanning microscopy (CLSM) as described [26]. A Leica TCS-SP8 confocal microscope fitted with a 20× multi-immersion objective lens (PL APO CS2 20×/0.75 IMM CORR HC; Leica Microsystems) was used for CLSM. Images were captured at 495–540 nm after excitation at 488 nm with a solid-state laser.

GUS staining was performed as described previously [6], except that the incubation time for GUS staining of ovules was changed from 0.5 to 1.5 h. Images were observed and captured by light microscope BX51 (Olympus, Tokyo, Japan) equipped with a DP70 cooled CCD camera (Olympus, Tokyo, Japan).

Silique images were captured and recorded using a SteREO Lumar V12 stereomicroscope (Carl Zeiss, Oberkochen, Germany).

Images were processed for publication using Adobe Photoshop CC (Adobe Systems Inc., San Jose, CA, USA).

### 4.4. Real-Time PCR Analysis

Total RNA was prepared from inflorescences using the RNeasy Plant Mini kit (Qiagen, Tokyo, Japan). First-strand DNA was synthesized from the RNA sample used for RNA-seq analysis using a ReverTra Ace qPCR RT kit (Toyobo, Osaka, Japan) according to the manufacturer’s protocol. The gene-specific primers used for real-time PCR are listed in Appendix A. Real-time PCR was performed using an Applied Biosystems 7300 Real-Time PCR System (Life Technologies, Tokyo, Japan) and THUNDERBIRD SYBR qPCR mix (Toyobo) with 40 cycles of denaturation at 95 °C for 15 s and extension at 60 °C for 1 min. ABI sequence detection software (version 1.4; Life Technologies, Tokyo, Japan) was used for quantification. *UBQ10* was used as an internal control to normalize the RNA quantity. Each sample was analyzed in triplicate.

### 4.5. AlphaFold Structure Prediction

The structures of AtGEX1 and OsGEX1, as predicted by AlphaFold2, were obtained from the AlphaFold Protein Structure Database [27]. To predict the structures of AtGEX1 and OsGEX1, which lacked a putative signal sequence by AlphaFold2 [20], we used the ColabFold Google Colab notebook [28]. Structural alignments were performed by The PyMOL Molecular Graphics System, Version 2.4.1 (Schrödinger, LLC, New York, NY, USA).

## Figures and Tables

**Figure 1 plants-11-01808-f001:**
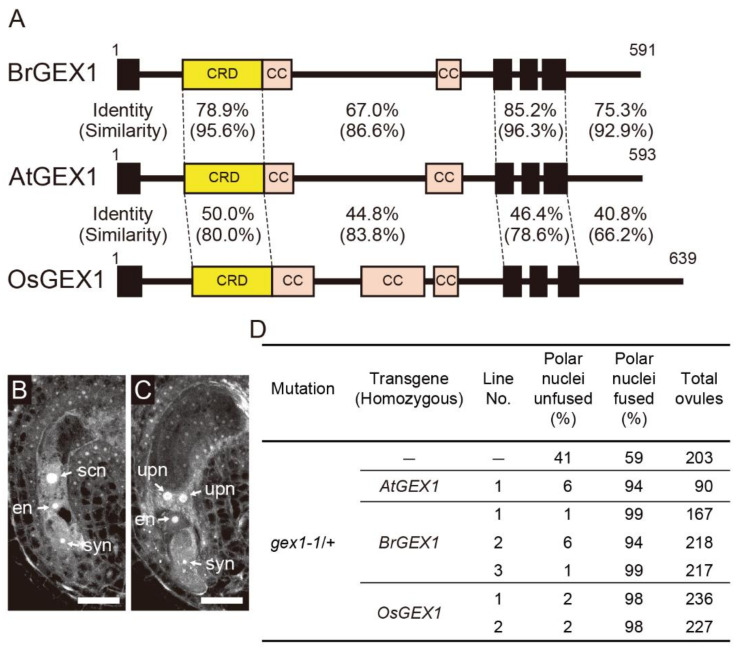
Expression of *Brassica rapa* or *Oryza sativa* GEX1 orthologs rescued the polar nuclear fusion defect of the *Arabidopsis gex1-1* mutant female gametophytes. (**A**) Schematic representation of AtGEX1, BrGEX1, and OsGEX1. GEX1 has 1 predicted N-terminal signal sequence (SS), 1 Cys-rich domain (CRD), 2 coiled-coil domains (CC), and 3 transmembrane domains (TMs). Amino acid sequence identities and similarities of each region of BrGEX1 and OsGEX1 between AtGEX1 are shown. (**B**,**C**) Confocal laser-scanning microscopy (CLSM) of the ovules of the *gex1-1*/+ plant. Half of the ovules are morphologically normal (**B**), and in the other half, the polar nuclei remained unfused (**C**). syn, synergid nucleus; en, egg nucleus; scn, secondary nucleus; upn, unfused polar nucleus. Bars = 20 µm. (**D**) Suppression of the polar nuclear fusion defect of the *gex1-1* female gametophytes by BrGEX1 or OsGEX1 expression. The ovules were analyzed by CLSM.

**Figure 2 plants-11-01808-f002:**
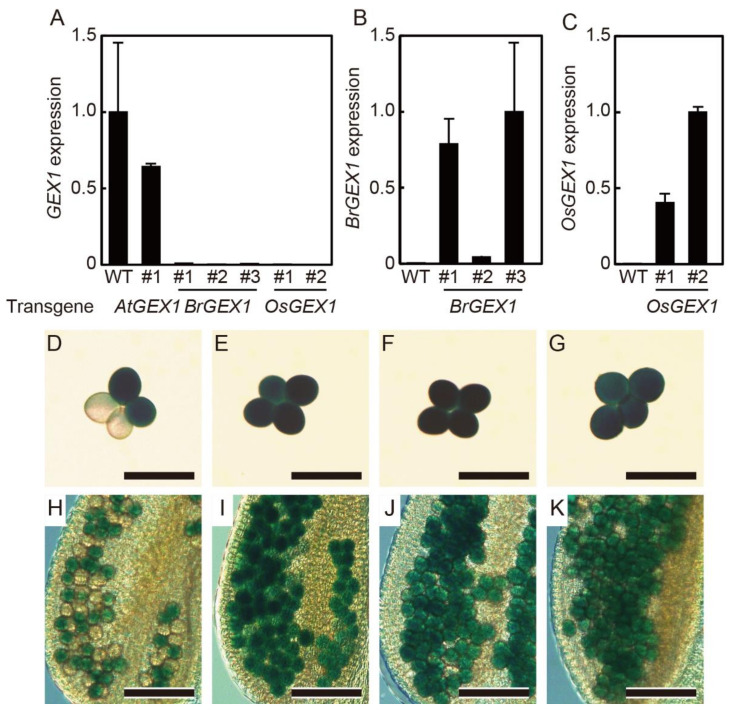
The *gex1-1*/*gex1-1* homozygous plants expressing AtGEX1, BrGEX1, or OsGEX1 from the *AtGEX1* promoter. (**A**) *AtGEX1* mRNA levels measured using qRT-PCR in flowers of wild-type (WT) and *gex1-1*/*gex1-1* plants homozygous for *pAtGEX1*: *AtGEX1*, *pAtGEX1*: *BrGEX1* or *pAtGEX1*: *OsGEX1*. (**B**) *BrGEX1* mRNA levels measured using qRT-PCR in flowers of wild-type (WT) and *gex1-1*/*gex1-1* plants homozygous for *pAtGEX1*: *BrGEX1*. (**C**) *OsGEX1* mRNA levels measured using qRT-PCR in flowers of wild-type (WT) and *gex1-1*/*gex1-1* plants homozygous for *pAtGEX1*: *OsGEX1*. (**D**–**K**) Images of GUS-stained pollen tetrads (**D**–**G**) and anther locules (**H**–**K**) from *gex1-1*/+ plants (**D**,**H**) or *gex1-1*/*gex1-1* plants homozygous for *pAtGEX1*: *AtGEX1* (**E**,**I**), *pAtGEX1*: *BrGEX1* (**F**,**J**) or *pAtGEX1*: *OsGEX1* (**G**,**K**). Bars = 50 µm (**D**–**G**), 100 µm (**H**–**K**).

**Figure 3 plants-11-01808-f003:**
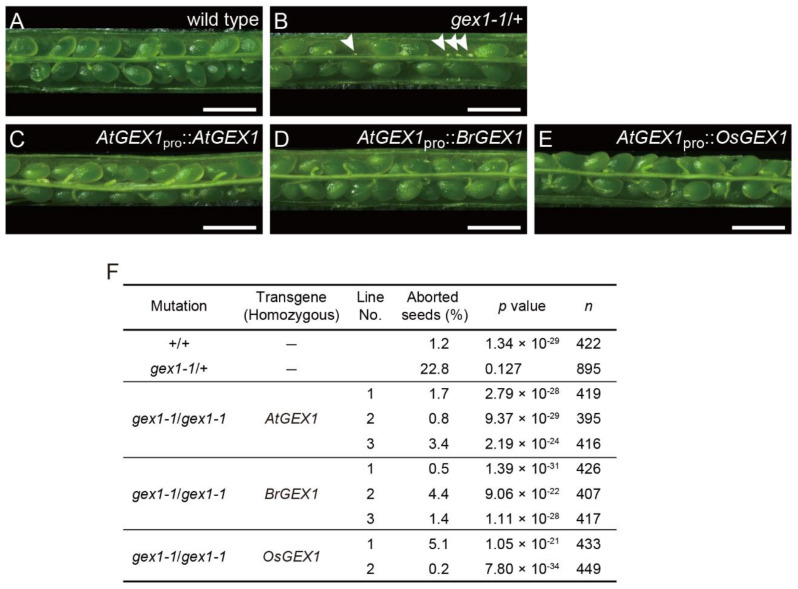
Expression of BrGEX1 or OsGEX1 from the *AtGEX1* promoter-rescued seed abortion of the *gex1-1*/*gex1-1* homozygous mutant. (**A**–**E**) Siliques of wild-type (**A**), *gex1-1*/+ (**B**) and *gex1-1*/*gex1-1* plants homozygous for *pAtGEX1*: *AtGEX1* (**C**), *pAtGEX1*: *BrGEX1* (**D**) or *pAtGEX1*: *OsGEX1* (**E**). *Arrowheads* in (**B**) show aborted seeds. Bars = 1 cm. (**F**) Percentages of aborted seeds in *gex1-1* mutants.

**Figure 4 plants-11-01808-f004:**
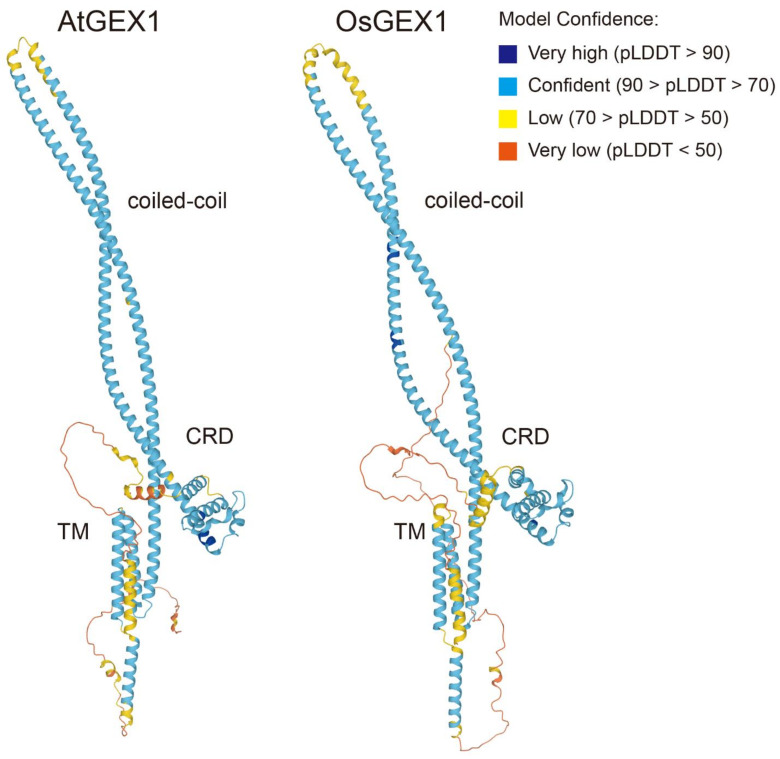
Model of AtGEX1 and OsGEX1 structures, as determined by the AlphaFold2 protein structure and folding prediction algorithm [20]. The structures for AtGEX1 (left: Q681K7) and OsGEX1 (right: Q67UU1) are shown. Chains are colored by the predicted Local Distance Difference Test (pLDDT) score (blue to red). CRD, a coiled-coil region, and transmembrane domains (TM) are shown.

## Data Availability

The authors confirm that the data supporting the findings of this study are available within the article and its Appendix A.

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
