# Peer review of "Expression of GEX1 Orthologs of *Brassica rapa* and *Oryza sativa* Rescued the Nuclear Fusion Defect of the Arabidopsis GEX1 Mutant"

_plants, 2022, doi:10.3390/plants11141808_

Round 1

Reviewer 1 Report

This manuscript addresses the important question of mechanisms of nuclear fusion during sexual reproduction. The work focuses on functional relationships between 3 plant members of the broadly conserved nuclear fusion protein family, KAR5/GEX1/Brambleberry. The essential findings are that both the Brassica rapa and Oryza sativa family members rescue polar nuclear fusion and sperm-egg nuclear fusion in Arabidopsis thalianaKAR5/GEX1/Brambleberry mutants. The experiments are well done, and the results are convincing. 

The authors make the case that, based on low oveall sequence identity, rescue is somewhat surprising. On the other hand, the sequence identity of what the authors term the N-terminal regions is higher than that of the full-length proteins. Furthermore, the manuscript reports that structures predicted by AlphaFold are quite similar. Given that evolution selects for function and that structure is central to function, it does not seem too surprising that these highly structurally similar proteins can substitute for each other. But the experiments needed to be done, and they do represent an advance for the field. Nuclear fusion driven by KAR5/GEX1/Brambleberry is central to sexual reproduction in organisms across taxa, and even though the first member of this protein family was discovered nearly 30 years ago, we still lack even a rudimentary understanding of mechanisms. Thus, it is exciting to see these authors taking first steps. 

The manuscript, however, must provide a larger perspective for the reader. As it stands, most readers would be completely unaware that the proteins worked on here are part of the larger KAR5/GEX1/Brambleberry family. The family is not mentioned by name. That other organisms use members of the family is vaguely alluded to, and the paper (ref. 14) first recognizing that these 3 proteins were members of the same family is cited, but the authors need to provide a more thorough and scholarly synopsis. Placing their work in this larger context will only help readers understand the broader importance and relevance of these findings across biology.

In keeping with conventions in model organisms, the text needs to use the term KAR5/GEX1/Brambleberry when the family is described in the Introduction. It might be argued by some that the proteins addressed in this paper should be named AtKAR5, BrKAR5, and OsKAR5. That might be too much of a jump, though, and it would be sufficient to use the terms AtGEX1, BrGEX1, and OsGEX1. The text already uses the latter 2 terms and refers to the ArabidopsisGEX1 simply as GEX1. Changing the name to AtGEX1 will make the text clearer to understand. Other comments are listed below:

Introduction is repetitive and is not unified: Lines 31-40 say essentially the same thing at least twice.

Line 74-75 (and see below): need to specify by residue numbers and in a new graphic, what is meant by N- and C-termini.

Lines 80-81: Question being asked is ambiguous and vague. In some places the text indicates that the question is whether the genes from other organisms can rescue function in At, and here the question being asked is about “functional differences in nuclear fusion.” Need to elaborate and to be consistent.

Lines 182-188: Text is confusing about identities. First sentence says identity is low, next says it’s high. Reader won’t understand. Need to show linear graphic of domains of the 3 GEX1s, including their predicted TMDs (Phobius or DeepTMHMM), in the main text along with precise indication of the segments defined as N-terminal domains.

Lines 194-195 and 243-244: Need to explain meaning of “luminal domain.” Which lumen is being referenced? The nuclear envelope is composed of two membranes. Do the authors have a model for the topology of the protein in the nuclear envelope? Text needs to specify which membrane is being referred to when the term nuclear membrane is used. This confusing sentence illustrates the importance of using more precise words: “This is in good contrast to the gex1-1 mutant female gametophytes, which were defective in nuclear membrane fusion following the fusion of the outer nuclear membranes [8].”

Lines 222-224 are confusing. I don’t understand the ideas related to HAP2/GCS1 being proposed here?

Need to provide root mean square deviations of other protein pairs to help the reader understand the importance of the 6.0 Å value for these proteins.

Reviewer 2 Report

Nicely written paper, conclusions justified, interesting contrast to the published results with HAP2 (i.e. not able to complement). 

A few grammatical corrections.

Line 29, pollen germinates, not pollens germinate (pollen is plural). 

line 32, central cell, not central cells (each embryo sac has only 1 central cell).  similar comment for line 34. 

line 69, N-terminus, not N-terminal

line 101, microscopy, not microscope

line 191, rephrase, not "showed the OsGEX1 to have a similar tertiary structure to that of GEX1", better would be "OsGEX1 had a similar tertiary.... 

Reviewer 3 Report

It is always important to confirm whether "conserved" genes are really functionally conserved, and this paper clearly showed that the "conserved" GEX1 from different species works in Arabidopsis. I have several suggestions to improve the manuscript.

1. I am curious to know why the authors did not do sequencing to confirm the genotype. GUS staining in Fig. 2 clearly shows it but usually, sequencing is the golden standard.

2. Figure 3. If you want to mention fertilization failure, this has to be checked. The HTR10 sperm chromatin marker can be used to check whether chromatin condensation (karyogamy is successful)  is present or not. 

3. Fig. 4. I strongly suggest showing Brassica data and comparisons as well.

4. The last sentence of the discussion is not clear. Do Rice and B. rapa perform inner membrane fusion upon fertilization? Then they should have the system with GEX1? What is the phenotype of gex1 mutant in rice and B. rapa? It is not clear what the author is trying to discuss here, I could not follow this part. Please clarify.
